# Aloe-Vision: Robust Vision-Language Models for Healthcare

**Jaume Guasch-Martí**                                    JAUME.GUASCH@BSC.ES
**Enrique Lopez-Cuena**                                  ENRIQUE.LOPEZ@BSC.ES
**Martín Suárez-Fernández**                            MARTIN.SUAREZ@BSC.ES
**Jordi Bayarri-Planas**                                  JORDI.BAYARRI@BSC.ES
**Anna Arias-Duart**                                  ANNA.ARIASDUART@BSC.ES
**Dario Garcia-Gasulla**                                  DARIO.GARCIA@BSC.ES
*Barcelona Supercomputing Center (BSC)*

**Editors:** Accepted for publication at MIDL 2026

## Abstract

Large Vision-Language Models (LVLMs) specialized in healthcare are emerging as a promising research direction due to their potential impact in clinical and biomedical applications. However, progress is constrained by the scarcity of high-quality medical multimodal data, concerns about robustness in safety-critical settings, and the narrow and potentially contaminated evaluation benchmarks that limit reliable assessment. To address these issues, the field requires state-of-the-art solutions to be fully open and reproducible systems in which all components can be inspected, evaluated, and improved. This work introduces **Aloe-Vision-Data**, a large-scale, quality-filtered mixture which integrates both medical and general domains across multimodal and text-only sources, designed for direct use in model fine-tuning. Building on this dataset, we train the **Aloe-Vision** family of medical LVLMs, openly released with full weights, training recipes and data, in two scales (7B and 72B). Through comprehensive benchmarking, we demonstrate that high quality training mixtures produce balanced LVLMs which yield significant gains over the baseline models without compromising general capabilities, achieving competitive performance with respect to state-of-the-art alternatives. To support reliable evaluation, we introduce **CareQA-Vision**, a carefully curated vision benchmark derived from MIR and EIR exams, the residency entrance exams for medical and nursing specialists in Spain, offering novel vision questions with low likelihood of contamination. Finally, we show that current LVLMs remain vulnerable to adversarial and misleading inputs, underscoring reliability challenges in clinical contexts.

**Keywords:** LVLMs, Healthcare, Adversarial evaluation

## 1. Introduction

Large Vision-Language Models (LVLMs) have achieved remarkable progress in general multimodal domains, and their ability to jointly process images and text makes them naturally aligned with the multimodal nature of medicine, where visual data (*e.g.*, X-rays, CT scans, histopathology slides) must be interpreted alongside clinical narratives, patient histories, and diagnostic reports. However, despite their strong potential, progress in medical LVLMs remains limited and still falls short of human-level performance (Sun et al., 2024), largely due to three key challenges. First, the availability of high-quality, medical image-text data remains severely limited, restricting both the scale and diversity required to train LVLMs

that generalize across modalities, pathologies, and anatomical structures. Second, evaluation practices rely heavily on medical VQA benchmarks that are often noisy and have been publicly available long enough to risk contamination, resulting in unreliable or overly optimistic assessments of model performance. Third, even state-of-the-art LVLMs exhibit notable vulnerabilities to adversarial, ambiguous, or misleading prompts, exposing robustness failures that are unacceptable in safety-critical clinical environments. Together, these limitations highlight an urgent need for fully open, well-documented, reproducible LVLMs that can serve as trustworthy foundations for clinical and biomedical applications.

To address these challenges, we introduce **Aloe-Vision**, a family of open and reproducible medical LVLMs achieving competitive performance with the state-of-the-art. The models are trained on **Aloe-Vision-Data**, a balanced and quality-filtered mixture explicitly curated to be directly usable for LVLM fine-tuning. It integrates four types of data: (1) multimodal medical datasets for visual clinical reasoning, (2) general multimodal data to preserve visual capabilities, (3) medical text-only data for domain-specific knowledge, and (4) general text-only data to maintain conversational fluency. The proportion of each data category is determined by the number of loss-contributing tokens rather than raw sample counts, ensuring uniform influence across data modalities and preventing longer samples from dominating the training signal. Beyond standard cleaning, we apply a semi-automatic filtering procedure that removes low-quality or inconsistent annotations using LVLM-based scoring and answer perplexity. We prevent leakage into benchmark datasets eliminating duplicates through perceptual hashing. To obtain a holistic view beyond standard medical VQA benchmarks, we assess model performance across medical multimodal, medical text-only, general multimodal, and general text-only tasks. We also incorporate a new medical vision benchmark, **CareQA-Vision**, curated to assess clinical reasoning on entirely unseen cases. Finally, we evaluate the robustness of medical LVLMs under adversarial conditions, testing their reliability when confronted with misleading or contradictory multimodal cues.

In summary, the core contribution of this work is delivering fully open and reproducible medical LVLMs that the community can build upon. This is enabled by the following three components, made publicly available[1]:

- **Aloe-Vision-Data**: a ready-to-train balanced training mixture across modality (multimodal vs. text-only) and domain (medical vs. general).

- Aloe-Vision: a family of open medical LVLMs with improved robustness against adversarial attacks and competitive performance in healthcare and general domains.

- **CareQA-Vision**: a benchmark curated from Spanish residency entrance exams for evaluating model performance on unseen data.

## 2. Related Work

**Medical Multimodal Models** The development of Large Language Models (LLMs) established a foundation for processing clinical text (Singhal et al., 2023b; Chen et al., 2023). This architecture was subsequently expanded into Multimodal LLMs by integrating vision encoders to enable the processing of mixed-modality inputs (Liu et al., 2023; Bai

---

1. https://huggingface.co/collections/HPAI-BSC/healthcare-vlms-aloe-vision

et al., 2023). These general advancements have been translated to the healthcare domain to handle medical imaging. Early adaptations such as MedFlamingo (Moor et al., 2023), LLaVA-Med (Li et al., 2023), and MedGemini (Saab et al., 2024) demonstrated that fine-tuning general LVLMs on medical images yielded strong visual-question-answering (VQA) abilities, yet they remained limited to modest parameter and data scales. Subsequent developments have employed more capable foundation models with substantially larger training corpora. HuatuoGPT-Vision (Chen et al., 2024) integrates clean PubMed VQA pairs into a Yi 1.5 (Young et al., 2024) based model, making it one of the few fully open efforts, although its performance now lags behind more recent systems. GMAI-VL (Li et al., 2024) couples a three-stage alignment pipeline with 5.5 M image-text pairs, but its model remains closed. Finally, the most recent Lingshu (Xu et al., 2025) and Hulu-Med (Jiang et al., 2025) improved the state-of-the-art but are only partially open, limiting transparency and evaluation. Our work introduces **Aloe-Vision**, which matches the performance of these state-of-the-art systems while offering fully reproducible medical LVLMs at 7B and 72B scales, released with complete training data, recipes, and preprocessing steps.

**Large-Scale Medical Multimodal Datasets** Training corpora have been scaled from thousands to millions of pairs. PubMedVision (Chen et al., 2024) filters and refines existing PubMed image collections (PMC-OA, LLaVA-Med PMC, PMC-Inline) yielding 915K medical images that generate 1.3 million VQA pairs via GPT-4V synthesis. GMAI-VL-5.5M (Li et al., 2024) aggregates 219 expert datasets across 13 imaging modalities and 18 specialties, converting classification and detection annotations into 5.5 million caption and instruction samples using GPT-4o; however, the dataset itself is not publicly released. MedTrinity-25M (Xie et al., 2024) employs retrieval-augmented generation with domain-specific segmentation models to auto-generate ROI-grounded triplets for 25 million images without requiring paired text descriptions. Recent works assemble four-way mixtures spanning medical multimodal, medical text, general multimodal, and general text: Lingshu (Xu et al., 2025) curates such a mixture but does not disclose category ratios while Hulu-Med (Jiang et al., 2025) reports a similar four-way composition but does not release the mixture. In contrast, we release the first **ready-to-train**, balanced mixture which uses a loss-token-based weighting scheme to avoid overfitting to longer samples.

**Evaluation Benchmarks** Traditional medical VQA datasets such as SLAKE (Liu et al., 2021), PathVQA (He et al., 2020), and VQA-RAD (Lau et al., 2018) remain widely used but are too limited to evaluate modern LVLMs effectively. This gap has motivated the development of more comprehensive and challenging benchmarks. GMAI-MMBench (Ye et al., 2024) consolidates 284 expert datasets into 26k questions spanning region-, box-, mask-, and image-level reasoning across 38 imaging modalities and 18 clinical departments. OmniMed-VQA (Hu et al., 2024) converts 73 medical classification datasets into 128k multiple-choice questions (MCQ) covering 12 modalities and over 20 anatomical regions. In pathology, PathMMU (Sun et al., 2024) provides 33k expert-validated QA pairs, demonstrating that LVLMs significantly underperform board-certified pathologists. ProbMed (Yan et al., 2025) introduces adversarial evaluation by pairing ground-truth queries with negated hallucination versions. We assemble a comprehensive benchmark suite that unifies medical multimodal, medical text-only, general multimodal, and general text-only tasks to obtain a holistic view of model quality. To do so, we combine the benchmarks described above, introduce

Table 1: Final composition of the SFT training mixture after preprocessing, leakage removal, quality filtering, and token-based rebalancing.

| Dataset | Samples | Loss tokens (M) | Modality | Domain | B. Boxes |
|---|---|---|---|---|---|
| PubMedVision (Chen et al., 2024) | 1.26M | 175.3 | MM | Medical | No |
| MedMax (Bansal et al., 2024) | 409K | 33.7 | MM | Medical | No |
| MeCoVQA (Huang et al., 2025) | 27.5K | 0.7 | MM | Medical | Yes |
| Med-GRIT (Huang et al., 2024) | 17.7K | 2.6 | MM | Medical | Yes |
| MedTrinity-25M (Xie et al., 2024) | 330K | 55.5 | MM | Medical | Yes |
| Cambrian-10M (Tong et al., 2024) | 668K | 65.4 | MM | General | No |
| Aloe (Gururajan et al., 2024) | 756K | 190.3 | Text | Medical | – |
| Magpie-Ultra-v1.0 (Team, 2024) | 100K | 116.6 | Text | General | – |
| **Total** | 3.57M | 640.0 | – | – | – |

**CareQA-Vision** as a contamination-free medical vision benchmark, and evaluate LVLMs under adversarial conditions using the HEART framework (Suárez-Fernández et al., 2026).

## 3. Training Data

To preserve broad conversational competence while improving medical visual reasoning, we construct a balanced training mixture along two axes: modality (multimodal vs. text-only) and domain (medical vs. general). Within medical multimodal data, both global image understanding and fine-grained, region-referenced reasoning are included. The final **Aloe-Vision-Data** mixture draws from eight datasets, as detailed in Table 1.

**Preprocessing and normalization.** Samples are converted to a unified conversational schema (alternating user/assistant messages), enabling multi-turn dialogue. This structure supports interleaved multimodal inputs, allowing for sequences containing multiple images mixed with text. Region-level supervision is standardized using Qwen2-VL (Wang et al., 2024) format, using the markers <|box_start|>$(x_{tl}, y_{tl})$, $(x_{br}, y_{br})$<|box_end|> and normalizing coordinates to $[0, 1000]$. Cleaning steps included (1) removal of missing/corrupted images, (2) a $50 \times 50$ minimum size, (3) capping at 5 images/sample to avoid training instability and (4) sequence-length filtering at 4096 tokens. MedTrinity-25M (Xie et al., 2024) is randomly subsampled to 400K sequences to avoid over-representation of images with rendered boxes.

**Evaluation leakage prevention.** Leakage into evaluation sets is explicitly controlled using 64-bit perceptual hashing (pHash) (Buchner, 2025) matching between all training and evaluation images, which detects near-duplicate images even under resizing or compression. This removes 6,273 training samples, ensuring reported evaluation gains are not overestimated by training-evaluation overlap with data used to fine-tune the models.

**Semi-automatic quality filtering.** Manual inspection of medical multimodal datasets revealed low-quality cases (*e.g.*, answers written on the image, image irrelevant to the question, mismatched question and answer, see Figure 4). Given the scale, a two-signal semi-automatic filter is adopted:

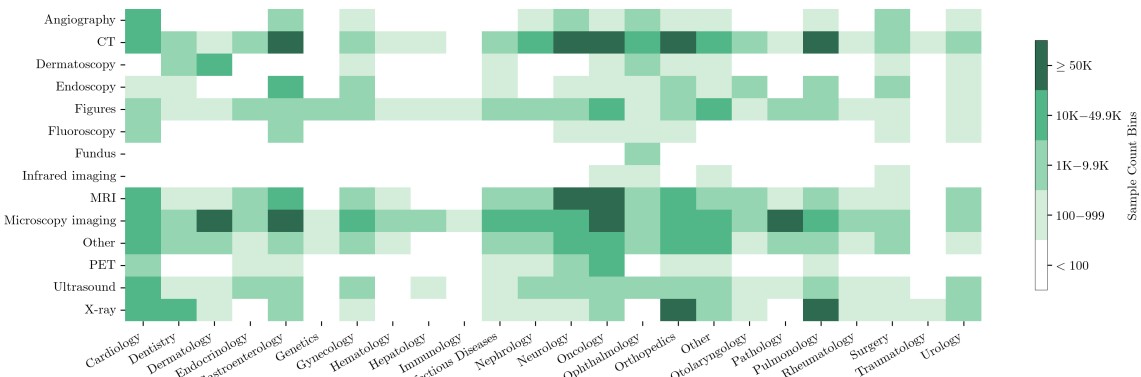

Figure 1: Category coverage analysis of the final training mixture across imaging modality (rows) and medical specialty (columns).

- **LVLM tagging.** Qwen2.5-VL-72B-Instruct (Yang et al., 2025) is prompted to produce a 1-5 quality score per sample based on coherence and relatedness between image, question, and answer. See an excerpt of the prompt in Appendix A.1.

- **Answer perplexity.** Qwen2-VL-7B-Instruct (Wang et al., 2024) is used to compute perplexity of the answer conditioned on image and question. Very low perplexity often flags trivial answers (e.g., answers visible in the image), whereas very high perplexity indicates noisy or incorrect annotations.

Thresholds for both quality scores and perplexity are manually defined per source by reviewing high- and low-score/perplexity examples, ensuring that filtering adapts to the specific characteristics of each dataset. In total, 541,237 samples are excluded.

**Token-based rebalancing.** Early training runs showed biases related to answer length even though sample counts were balanced. To correct this, mixtures are rebalanced by *loss-contributing tokens* (assistant tokens only), rather than by examples. For each dataset, token statistics (total, loss, text, image) are computed and sources with long answers, such as Chain-of-Thought reasoning traces, are subsampled to equalize their effective gradient contribution. This procedure preserves the intended modality/domain proportions while mitigating bias from long-form datasets.

**Coverage analysis** We assess dataset diversity across three axes (image modality, medical specialty, and anatomical structure), focusing on the coverage of their combinations rather than balance along each individual axis. During the semi-automatic quality filtering (Section 3), Qwen2.5-VL-72B-Instruct is additionally prompted to tag each sample with categories along these axes using the information provided by the triplet (question, image, answer), enabling construction of coverage heatmaps. Figure 1 illustrates the *image modality* vs. *medical specialty* distribution. After excluding nonsensical cases (e.g., *fundus-bones*, *angiography-dentistry*), the dataset exhibits strong and balanced representation.

Table 2: Evaluation suite. MCQ = multiple-choice; Y/N = yes/no; OE = open-ended (J = LLM-as-judge).

| Benchmark | Modality | Domain | Task | Samples |
|---|---|---|---|---|
| PathMMU (Sun et al., 2024) | Multi | Medical | MCQ | 1.1K |
| GMAI-MMBench (Ye et al., 2024) | Multi | Medical | MCQ | 4.5K |
| OmniMedVQA (Hu et al., 2024) | Multi | Medical | MCQ | 89K |
| ProbMed (Yan et al., 2025) | Multi | Medical | Y/N | 57K |
| SLAKE (Liu et al., 2021) | Multi | Medical | OE (J) | 2K |
| MMMU (Yue et al., 2024) | Multi | General | MCQ | 1.4K |
| MultiMedQA (Singhal et al., 2023a) | Text | Medical | MCQ | 7K |
| MMLU (Hendrycks et al., 2021) | Text | General | MCQ | 14K |

**Final mixture.** The final SFT mixture contains ∼**3.57M samples** and ∼**640M loss tokens** (see Table 1). By loss tokens, the allocation is: *medical multimodal* 41.8% (267.8M), *medical text-only* 29.7% (190.3M), *general text-only* 18.2% (116.6M), and *general multimodal* 10.2% (65.4M). Overall, multimodal data contributes 52% of loss tokens and medical data represents 71.5%.

## 4. Evaluation data

We evaluate models using a comprehensive benchmark suite that spans medical multimodal, medical text-only, general multimodal, and general text-only tasks, providing a unified and reproducible assessment of overall model quality. We standardize execution using `VLMEvalKit` (Duan et al., 2024) and `lm-eval-harness` (Gao et al., 2024) to ensure consistent and reproducible evaluation across models. Prior studies focus on a limited subset of medical VQA benchmarks (*e.g.*, PathVQA, VQA-RAD, SLAKE), which not only narrow the evaluation scope but also risk data leakage, as many of these datasets have been publicly available for years. After manual inspection, we exclude PathVQA and VQA-RAD due to quality concerns (*e.g.*, incorrect references, image-independent questions) and adopt newer, higher-fidelity benchmarks that better capture clinical diversity and reasoning competence. The final benchmark suite is summarized in Table 2.

**CareQA-Vision.** To ensure a contamination-free evaluation, we curate an extension of the CareQA dataset (Arias-Duart et al., 2025). CareQA-Vision is derived from MIR and EIR exams, the residency entrance exams for medical and nursing specialists in Spain, where all questions are curated by medical experts. It incorporates vision-based questions in nursing and medicine from 2020 to 2024 exams. Originally in Spanish and presented in a multiple-choice format, the questions are translated to English and converted into open-ended format using Qwen2.5-72B-Instruct (Yang et al., 2025), followed by manual verification. Although some images include Spanish words, they are not required to solve the tasks. CareQA-Vision contains a total of 301 questions: 70 from nursing and 231 from medicine. Closed questions represent 60% of nursing and 53.2% of medicine queries, with the remaining 136 questions being open-ended. Figure 2 shows both open-ended and close-ended examples. Despite its relatively limited size, this dataset provides a meaningful

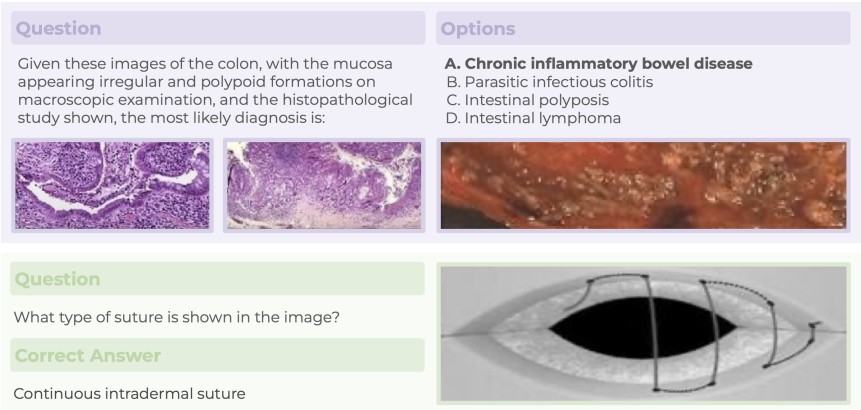

Figure 2: CareQA-Vision examples. Top: a medical MCQ, with the correct option in **bold**. Bottom: a nursing question in an open-ended format.

indicator of model performance in the healthcare domain, as it consists of high-quality, expert-reviewed questions with a low risk of training-set contamination.

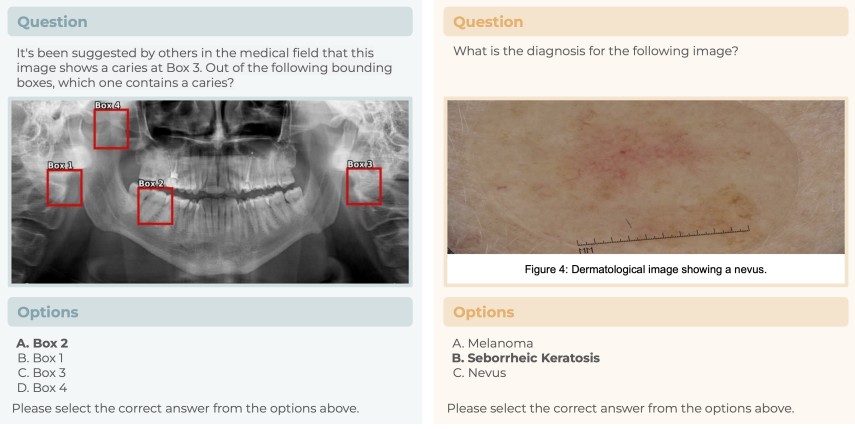

Figure 3: Adversarial examples, correct option in **bold**. Left: sycophancy example based on a detection task. Right: caption example for a classification task.

**Adversarial Benchmark.** To assess the robustness of state-of-the-art LVLMs under misleading conditions, we evaluate models using the HEART adversarial benchmark (Suárez-Fernández et al., 2026). Constructed from eight existing medical datasets spanning multiple imaging modalities (*e.g.*, X-ray, MRI, ultrasound), it includes both classification and detection tasks (see Figure 3, left/right) and introduces four types of adversarial attacks designed to probe whether model predictions remain grounded in the visual evidence. These include: (1) Sycophancy, where suggested labels or bounding boxes are inserted into the prompt, (2) Captions, where incorrect captions are embedded directly within the image, (3) Prompt,

where incorrect captions are provided in the text prompt, and (4) Legends, where mismatched legends accompany the image. In total, the benchmark contains 24,311 samples, out of which 2,979 represent a baseline that serves for comparison against each of the adversarial attack types. These settings test whether models can accurately interpret the underlying visual content despite the presence of misleading distractors.

## 5. Experiments

We evaluate *Aloe-Vision-7B* and *Aloe-Vision-72B* against state-of-the-art LVLMs using the benchmark suite described in Section 4. In addition to standard performance comparisons, we report results on the adversarial robustness evaluation. Comprehensive ablation studies are provided in Appendix D, covering the effect of (1) training-mixture composition, (2) evaluation-leakage control, and (3) semi-automatic quality filtering.

### 5.1. Setup

All experiments use single-stage supervised fine-tuning (SFT) with `TRL` (von Werra et al., 2020) library on alternating `user`/`assistant` dialogues, optimizing next-token cross-entropy over assistant turns. We fine-tune the Qwen2-VL-Instruct family (Wang et al., 2024) at 7B and 72B parameters, retaining the native multi-image interface and the Qwen-style chat template. The resulting models are referred to as *Aloe-Vision-7B* and *Aloe-Vision-72B*. Compute runs were performed in two European HPC systems: Leonardo (CINECA, A100 64GB ×4 per node) and MareNostrum 5 ACC (BSC, H100 64GB ×4 per node). The full training configurations for both models are summarized in Appendix B.

**Models.** We benchmark our models against a set of open-source LVLMs, categorized into: (1) General-purpose LVLMs, specifically the Qwen2-VL (Wang et al., 2024) and Qwen3-VL (Bai et al., 2025) families, and Kimi-VL (Team et al., 2025), which serve as strong reference for general visual capabilities; and (2) Specialized Medical LVLMs, including HuatuoGPT-Vision (Chen et al., 2024), Lingshu (Xu et al., 2025), Hulu-Med (Jiang et al., 2025) and Chiron-o1 (Sun et al., 2025). Finally, we include MiMo-VL (Yue et al., 2025) and GLM-4.5V (Hong et al., 2025), representing the emerging class of *reasoning* models trained to use extended thought processes for complex visual problems. In addition, we include GPT-5.2 (OpenAI, 2025) as a closed-source baseline to contextualize the performance of open and reproducible medical LVLMs against state-of-the-art proprietary systems.

**Adversarially Robust Aloe-Vision.** To mitigate the impact of adversarial attacks, we develop an adversarially robust (AR) variant of Aloe-Vision by introducing an additional post-training SFT stage using adversarial samples. The adversarial training set is created by applying all attack types described above **exclusively to a single imaging modality** (FracAtlas (Abedeen et al., 2023)), which is not included in the HEART benchmark. The purpose of this design is to test whether robustness learned from one modality can generalize to others (*i.e.*, whether fine-tuning on adversarial examples from a single source domain yields cross-modal and cross-specialty robustness). This second stage consists of a single SFT epoch over 17.2k samples.

**Evaluation protocol.** All multimodal benchmarks are implemented and run within VLM-EvalKit (Duan et al., 2024), and all text-only benchmarks are evaluated with *lm-evaluation-harness*. For every model, the benchmarks are run under identical settings to ensure fair and reproducible comparison. For multimodal tasks, inference uses greedy decoding, and accuracy is computed via exact string match for multiple-choice and yes/no formats. Open-ended answers in SLAKE and CareQA-Vision are scored with a majority voting LLM-as-judge protocol, where Qwen2.5-VL-72B (Yang et al., 2025), Llama-3.3-70B (Grattafiori et al., 2024) and Olmo-3-32B (Olmo et al., 2025) assign one of three rubric-based scores {0.0, 0.5, 1.0}. For text-only tasks, prediction followed the standard multiple-choice evaluation in `lm-evaluation-harness`, selecting the option with the highest log-likelihood as the model's answer.

**Human Evaluation.** Before reporting quantitative results, we assess the reliability of the judges (Qwen2.5-72B-Instruct (Yang et al., 2025), Llama-3.3-70B (Grattafiori et al., 2024) and Olmo-3-32B (Olmo et al., 2025)) used in open-ended benchmarks. To do so, we sample the open-ended medical subset of CareQA-Vision and ask experts to evaluate Aloe-Vision-72B-AR's answers following the same criteria as the LLM judge. For each question, experts were shown the correct answer (from the original exam) and the model's answer, and were asked to classify the model output as *correct*, *partially correct*, or *incorrect* (see Appendix E.2 for details). Inter-evaluator agreement among human experts, computed independently of the LLM judges, is moderate (Krippendorff's $\alpha = 0.796$) (Krippendorff, 2011), reflecting the inherent difficulty and subjectivity of the task. We then assess agreement when incorporating LLM-based judging by treating the automatic evaluation as the majority vote of the three independent LLM judges, and comparing this aggregated decision against the human annotations. Under this setup, overall agreement increases to $\alpha = 0.812$, indicating that the LLM judges are largely consistent with human expert assessments. Based on these findings, we consider the judge's evaluation sufficiently reliable, while acknowledging potential biases, such as a tendency to favor longer answers.

## 5.2. Results

**Evaluation Results** Table 3 summarizes the evaluation results, covering both general-purpose LVLMs (highlighted in gray) and medical LVLMs. Overall, the results reveal a clear separation between models trained with domain-specific medical data and general-purpose counterparts, as well as a consistent scaling trend with increasing model size. A notable exception is observed on the SLAKE benchmark, where Lingshu and Hulu-Med outperform the remaining models by approximately 15–20%. This gap is largely explained by the inclusion of the SLAKE training split in their respective training mixtures, exposing these models to highly similar samples. In contrast, Aloe-Vision models explicitly exclude all SLAKE examples from training, resulting in what we consider to be a more conservative and fair estimate of generalization performance on this benchmark.

**Small models (<10B).** Within the small-scale parameter group, Hulu-Med-7B achieves the strongest overall results, ranking first on four out of the ten benchmarks. The remaining datasets are led by Qwen3-VL-8B-Instruct, Aloe-Vision-7B-AR, and Lingshu-7B. At this model scale, the results suggest that architectural and pretraining advances in newer base

Table 3: Performance comparison across models of different sizes. Rows highlighted in gray denote general models, while the others are domain-specific.

| Model | CareQA-V MCQ | CareQA-V VQA | OMVQA | GMAI | PathMMU | ProbMed | SLAKE | MMMU | Multimed QA | MMLU |
|---|---|---|---|---|---|---|---|---|---|---|
| GPT-5.2 | 80.61 | 74.63 | 67.74 | 57.98 | 68.56 | 78.99 | 75.14 | 62.55 | 81.97 | 86.55 |
| **Small models (<10B)** | | | | | | | | | | |
| Qwen2-VL-7B-Instruct | 51.52 | 26.47 | 70.44 | 46.42 | 55.08 | 72.96 | 64.42 | 50.22 | 59.67 | 67.82 |
| Qwen3-VL-8B-Instruct | 58.79 | 38.24 | 77.78 | **54.40** | 56.92 | 78.51 | 71.77 | **63.22** | 66.27 | **74.95** |
| MiMo-VL-7B-RL | 59.39 | 28.31 | 62.22 | 44.33 | 53.77 | 74.50 | 61.83 | 54.89 | 55.88 | 68.42 |
| Chiron-o1-8B | 47.27 | 21.32 | 72.10 | 41.41 | 55.87 | 73.72 | 69.27 | 43.78 | 59.65 | 71.56 |
| Lingshu-7B | 56.97 | 31.25 | 82.05 | 52.31 | **66.55** | 78.92 | 78.51 | 57.33 | 62.09 | 69.37 |
| HuatuoGPT-Vision-7B | 50.30 | 16.91 | 71.91 | 49.36 | 57.36 | 76.14 | 57.40 | 41.78 | 57.93 | 67.61 |
| Hulu-Med-7B | 59.39 | **43.38** | **83.73** | 53.58 | 65.41 | **79.71** | **83.32** | 48.33 | **70.97** | 68.34 |
| Aloe-Vision-7B | 56.36 | 36.76 | 75.87 | 52.81 | 62.08 | 76.47 | 63.62 | 44.44 | 58.48 | 65.95 |
| Aloe-Vision-7B-AR | **60.61** | 36.40 | 77.10 | 53.96 | 65.32 | 78.98 | 63.48 | 48.56 | 61.82 | 66.31 |
| **Large models (>10B)** | | | | | | | | | | |
| Qwen2-VL-72B-Instruct | 72.73 | 45.59 | 76.09 | 51.03 | 64.71 | 74.47 | 67.58 | 61.11 | 74.25 | 81.86 |
| Qwen3-VL-30B-A3B-Instruct | 71.52 | 43.75 | 78.47 | 57.34 | 62.26 | 78.68 | 75.12 | 64.56 | 73.91 | 80.89 |
| Kimi-VL-A3B-Instruct | 55.15 | 40.44 | 73.23 | 48.44 | 55.34 | 78.66 | 63.95 | 54.44 | 59.21 | 69.04 |
| GLM-4.5V | 72.73 | **64.71** | 73.24 | 53.74 | 65.67 | 77.43 | 68.90 | **73.11** | 67.47 | 81.09 |
| HuatuoGPT-Vision-34B | 54.55 | 24.26 | 67.46 | 48.33 | 54.90 | 70.88 | 59.14 | 42.11 | 60.57 | 72.80 |
| Lingshu-32B | 64.85 | 42.28 | 83.45 | 53.54 | 67.60 | 80.84 | 87.28 | 59.78 | 72.08 | 81.32 |
| Hulu-Med-32B | 63.64 | 48.53 | **84.92** | **58.04** | 69.88 | **81.93** | 85.49 | 56.89 | 75.83 | 80.04 |
| Aloe-Vision-72B | **77.58** | 49.63 | 81.77 | 54.79 | **71.45** | 77.71 | 67.72 | 63.22 | 76.24 | **82.55** |
| Aloe-Vision-72B-AR | 75.76 | 48.53 | 83.97 | 55.12 | 70.32 | 77.86 | 68.99 | 62.44 | **76.35** | 82.52 |

models significantly improve performance, with Qwen3-VL consistently outperforming its Qwen2-VL counterparts across benchmarks.

**Large models (20B-106B).** Increasing model size yields consistent performance improvements across all evaluated benchmarks. To the best of our knowledge, there are currently no other open-source medical LVLMs at the 70B parameter scale. As a result, Aloe-Vision-72B is compared against the strongest available medical models, which are limited to the 32B–34B range. Among all evaluated models, Aloe-Vision-72B-AR achieves the strongest overall performance, combining high accuracy on general and text-only benchmarks with robust results on medical-specific datasets. Interestingly, despite its substantially smaller parameter count, Hulu-Med-32B achieves comparable performance and in some benchmarks it even surpasses Aloe-Vision-72B, highlighting the impact of architectural choices, training data scale, and training strategy beyond model size alone.

**Reasoning-oriented models.** Finally, we observe that models explicitly post-trained to generate intermediate reasoning traces before producing a final answer (GLM-4.5V, MiMo-VL-7B-RL, and Chiron-o1-8B) do not consistently outperform other models across the benchmark suite. Among these, only GLM-4.5V achieves top performance on two benchmarks, an advantage that may be primarily attributable to its substantially larger scale (106B parameters). Overall, these findings suggest that the majority of current medical vision-language benchmarks are predominantly knowledge-based and can be effectively addressed without multi-step reasoning. This observation raises an important open question regarding the practical utility of reasoning-oriented models in the medical vision domain

Table 4: Accuracies for the adversarial experiments. Models highlighted in gray are general-purpose, while the others are domain-specific. The best result in each column is in **bold**, the second-best underlined.

| Model | Classification | | | | Detection | | | | |
|---|---|---|---|---|---|---|---|---|---|
| | Base | Cap | Pmt | Syc | Base | Cap | Pmt | Syc | Leg |
| Random | 41.67 | | 41.67 | | 25.00 | | 25.00 | | |
| GPT-5.2 | 70.2 | 6.4 | 10.9 | 38.5 | 70.9 | 14.3 | 5.3 | 9.7 | 8.7 |
| GLM_4.5V | 70.2 | 0.3 | 8.6 | 21.0 | 80.0 | 3.8 | 12.6 | 20.2 | 14.4 |
| Kimi-VL-A3B-Instruct | 55.6 | 0.0 | 0.6 | 11.5 | 75.8 | 4.2 | 7.4 | 10.0 | 21.1 |
| MiMo-VL-7B | 54.6 | 0.3 | 2.2 | 6.4 | 63.9 | 0.9 | 1.5 | 8.9 | 2.3 |
| Qwen2-VL-7B | 51.6 | 0.2 | 2.4 | 10.1 | 65.1 | 16.5 | 8.5 | 17.6 | 22.6 |
| Qwen2-VL-72B | 59.4 | 1.0 | 5.8 | 35.6 | 78.4 | 6.3 | 5.7 | 20.5 | 8.7 |
| Qwen3-VL-8B-Instruct | 67.6 | 0.3 | 8.6 | 28.2 | 81.0 | 19.5 | 19.7 | 35.7 | 16.4 |
| Qwen3-VL-30B-A3B-Instruct | 66.8 | 0.1 | 13.2 | 28.9 | 81.0 | 20.8 | 37.1 | 34.3 | 26.2 |
| Chiron-o1-8B | 48.9 | 5.8 | 8.6 | 54.6 | 57.4 | 19.2 | 8.6 | 33.3 | 24.4 |
| HuatuoGPT-Vision-7B | 57.3 | 21.4 | 7.2 | 30.7 | 51.9 | 25.5 | 0.7 | 5.2 | 41.6 |
| HuatuoGPT-Vision-34B | 59.1 | **22.7** | 10.1 | 14.9 | 55.9 | 25.6 | 2.3 | 6.3 | 36.6 |
| Lingshu-7B | **78.8** | 1.2 | 18.8 | 45.4 | 78.1 | 4.7 | 8.6 | 22.3 | 17.8 |
| Lingshu-32B | 68.9 | 2.7 | 29.8 | 56.9 | 77.7 | 7.8 | 12.6 | 34.8 | 22.7 |
| Hulu-Med-7B | 61.2 | 1.3 | 0.0 | 18.6 | 68.9 | 1.8 | 0.1 | 5.0 | 26.3 |
| Hulu-Med-32B | 68.4 | 1.3 | 4.1 | 35.4 | 73.3 | 3.0 | 8.2 | 21.8 | 15.3 |
| Aloe-Vision-7B-S1 | 59.6 | 2.3 | 15.0 | 41.6 | 69.1 | 66.4 | 10.3 | 26.3 | 34.4 |
| Aloe-Vision-7B-S2 | 65.3 | 13.7 | 43.8 | 50.2 | 76.9 | **75.3** | **58.5** | **68.6** | **58.0** |
| Aloe-Vision-72B-S1 | 63.0 | 3.5 | 8.2 | **75.0** | 72.5 | 4.0 | 2.9 | 19.7 | 9.4 |
| Aloe-Vision-72B-S2 | 66.6 | 16.2 | **51.0** | 63.5 | **82.0** | 31.0 | 57.0 | 54.3 | 33.1 |

and highlights the need for dedicated benchmarks that require and evaluate structured reasoning capabilities.

**Closed-source baseline.** GPT-5.2 provides a useful reference point for the current performance level of proprietary multimodal systems. It achieves the strongest results on CareQA-V MCQ, CareQA-V VQA, MultimedQA, and MMLU, indicating particularly strong performance on text-only benchmarks. At the same time, several open models surpass GPT-5.2 on OMVQA, GMAI, PathMMU, ProbMed, SLAKE, and MMMU, showing that targeted domain adaptation can substantially narrow, and in some cases overcome, the gap to proprietary systems on medical vision-language evaluation. However, we view GPT-5.2 primarily as a contextual baseline rather than a directly comparable model, since the evaluated open-source baselines are standalone models and GPT-5.2 is a full system that may incorporate capabilities extending beyond the underlying model alone.

**Generalization and Robustness.** Both Aloe-Vision models perform particularly well on the CareQA-Vision benchmarks, indicating generalization to unseen medical distributions. Furthermore, Aloe-Vision and Aloe-Vision-AR exhibit nearly identical performance across all standard (non-adversarial) benchmarks, regardless of model size. This result indicates that the additional supervised fine-tuning stage introduced to improve robustness does not come at the cost of reduced accuracy on conventional evaluation settings.

**Adversarial Results** While previous results show performance on standard benchmarks, we next assess model robustness under more challenging adversarial conditions. As shown in Table 4, strong performance on standard evaluations does not necessarily imply robustness when models face ambiguous or adversarial inputs. These results are obtained using the adversarial datasets described in previous section (see Figure 3). First column (*Base*) shows the results without any modifications, while the remaining columns correspond to evaluations with different adversarial attacks. To differentiate the effect of attacks on global versus region-specific predictions, we report results separately for classification tasks (using image-level labels) and detection tasks (using bounding-box-level labels).

Most models show substantial degradation when misleading information is introduced, indicating that high baseline accuracy (*e.g.*, Hulu-Med or Lingshu) does not necessarily guarantee robustness. Among the different adversarial subsets, misleading captions embedded in the image is the most damaging strategy, with nearly all models collapsing under this condition, especially in the classification tasks (*e.g.*, Qwen2-VL-7B-Instruct drops from 51.6 to 0.2, and Lingshu-7B from 78.8 to 1.2). Regarding detection and classification tasks, detection generally proves more resilient than classification, suggesting that spatial grounding offers partial protection against textual manipulation.

As expected, Aloe-AR models consistently outperform their standard counterparts across the adversarial settings, confirming that explicit robustness training mitigates susceptibility to misleading or sycophantic cues. Furthermore, this confirms that robustness learned from FracAtlas transfers reliably to unseen medical specialties and image modalities.

Notably, GPT-5.2 shows significant performance drops under adversarial conditions, especially in caption-based attacks, indicating that even advanced proprietary models remain susceptible to misleading multimodal inputs.

## 6. Conclusion

We introduced **Aloe-Vision-Data**, a large-scale, quality-filtered and token-balanced instruction mixture spanning medical and general domains across multimodal and text-only sources, together with a fully reproducible training and evaluation pipeline for medical LVLM fine-tuning. Building on this foundation, we released the **Aloe-Vision** model family at 7B and 72B scales and demonstrated competitive performance across a broad benchmark suite that jointly measures medical multimodal, medical text-only, and general capabilities. To support reliable assessment, we proposed **CareQA-Vision**, a contamination-resistant benchmark derived from expert-curated Spanish residency exams, enabling evaluation on previously unseen medical cases. Finally, our adversarial analysis shows that strong standard-benchmark performance does not guarantee reliability under misleading inputs, and that targeted robustness fine-tuning improves resistance to such attacks without degrading conventional accuracy. Together, these resources provide a fully open and reproducible foundation for advancing trustworthy medical vision-language modeling. Building on this foundation, an important direction for future research is to systematically disentangle the relative contributions of model scale, data scale and quality, and training curricula, in order to better characterize the performance trade-offs in medical vision-language models.

## Acknowledgments

Anna Arias-Duart, Jordi Bayarri-Planas, and Jaume Guasch-Martí acknowledge their AI4S fellowship within the "Generación D" initiative by Red.es, Ministerio para la Transformación Digital y de la Función Pública, for talent attraction (C005/24-ED CV1), funded by NextGenerationEU through PRTR. This work was also partially supported by the EL-LIOT project funded by the European Union under Grant Agreement No. 10121439. We also acknowledge the computational resources provided by CINECA and the Barcelona Supercomputing Center (BSC). We are particularly grateful to the Operations department at BSC for their technical support. Finally, we would like to thank all the healthcare experts who participated, especially Anabel Antolínez Dueñas, Marina Arias Duart, Jordi Farguell Piulachs, Elena Toledo, and Andreu Pacheco Agustí for their time and expertise.

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

## Appendix A. Quality Filtering

Before supervision fine-tuning, we apply a semi-automatic quality filtering pipeline to remove incoherent or low-value samples from the training pool, as explained in Section 3. Figure 4 illustrates typical failure modes captured by this process.

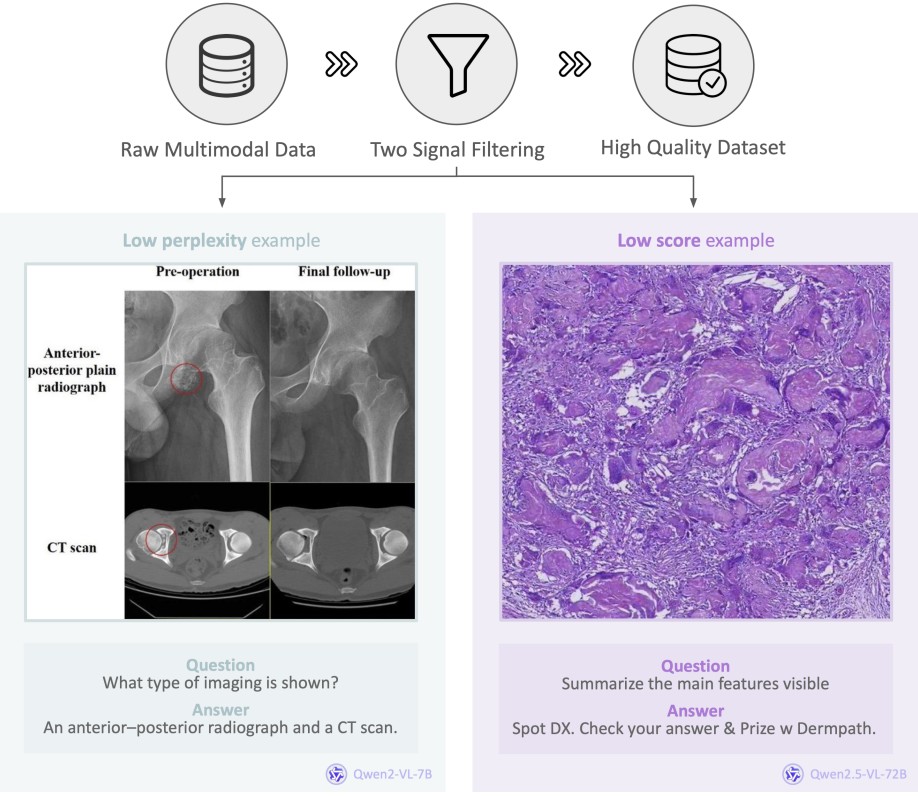

Figure 4: Semi-automatic quality filtering process. Below are examples of low-quality samples identified during filtering. Left: answer appears in the image (low score, low perplexity). Right: answer unrelated to the image (low score, high perplexity).

### A.1. Tagging Template

The following prompt is used with Qwen2.5-VL-72B-Instruct (Yang et al., 2025). We utilize a structured template that defines the task and output format, followed by a constrained taxonomy of tags.

**System Prompt: Medical Sample Extraction**

You are an expert medical assistant designed to categorize samples in different ways. A sample is composed of an optional image (or list of images), a question (or list of questions), and an answer (or list of answers).

Determine for each of the following sections which of the possible tags provided best describe the sample. For each category, you must only use the tags explicitly listed. If none apply, use the *Other* tag.

*[...Taxonomy definitions omitted for brevity, see Table 5...]*

Taking into account that for each category type, you must provide the tags in priority order, respond using the following format:
MOD[Modality] MED[Medical Field] ST[Sample Type] SBP[Specific Body Part]
GBP[General Body Part] SQ[Sample Quality]

**Example:**
*Image:* {CT scan}
*Question:* Is there evidence of a fracture in the distal radius?
*Answer:* Yes
*Response:*    MOD[X-ray] MED[Orthopedics] ST[Abnormality Detection] SBP[Bones]
GBP[Upper limbs] SQ[4]

*[Additional examples omitted for brevity]*

## A.2. Taxonomy Definitions

Table 5 details the five categorization dimensions and their vocabularies.

| Dimension | Constrained Tags |
|---|---|
| **Modality** | X-ray, CT, MRI, Ultrasound, PET, SPECT, Microscopy, Dermatoscopy, Fundus, OCT, Endoscopy, Fluoroscopy, Angiography, Infrared, Figures (Graphs/Charts), Other |
| **Medical Fields** | Cardiology, Neurology, Oncology, Orthopedics, Gastroenterology, Pulmonology, Dermatology, Ophthalmology, Pathology, Dentistry, Obstetrics & Gyn., Endocrinology, Hematology, Nephrology, Surgery, Infectious Diseases, Other |
| **Sample Types** | Diagnosis, Abnormality Detection, Modality ID, Anatomy ID, Comparison, Procedural Explanation, Visual Attributes, Severity Est., Prognosis, Treatment Suggestion, Etiology, Image Description, Result Analysis, Other |
| **Specific Body Parts** | Cell, Brain, Lungs, Heart, Stomach, Intestines, Liver, Pancreas, Spleen, Kidneys, Spine, Pelvis, Bones, Skin, Eyes, Teeth, Blood Vessels, Muscles, Joints, Breasts, Ears, Nose, Throat, Reproductive Organs, Non-body part, Other |
| **General Body Parts** | Head, Neck, Upper limbs, Lower limbs, Abdomen, Thorax, Pelvis, Non-body part, Other |
| **Sample Quality** | Score (1–5) evaluating coherence among image, question, and answer. |

Table 5: Taxonomy used for metadata extraction.

## Appendix B. Training

All training configurations are summarized in Table 6.

Table 6: Training configuration for Aloe-Vision-7B and Aloe-Vision-72B.

| Parameter | 7B | 72B |
|---|---|---|
| Stage | Single-stage full SFT | |
| Precision | BF16 | |
| Max. Sequence length | 4096 | |
| Epochs | 1 | |
| LR schedule | Cosine | |
| Gradient checkpointing | Enabled | |
| Parallelism | DeepSpeed ZeRO-3 | |
| Warmup | 3% | |
| Global batch size | 1024 | 2000 |
| Micro-batch size | 16 | 4 |
| Gradient accumulation | 2 | 5 |
| Optimizer | AdamW | AdamW 8-bit |
| Max. Learning rate | 3.75e-5 | 1.25e-5 |
| Total GPUs | 32 | 100 |
| Estimated GPU-hours | 500 | 4500 |

## Appendix C. Evaluation

More details on the diverse suite of medical and general benchmarks and the protocol used to ensure fair comparison and reproducibility are provided here.

**Benchmarks** Our medical multimodal category comprises **CareQA-Vision** (our contaminationfree benchmark derived from Spanish national medical and nursing exams, covering both MCQ and open-ended questions), **PathMMU** (Sun et al., 2024) (expert-validated pathology QA), **GMAI-MMBench** (Ye et al., 2024) (26k MCQs spanning image/box/ mask/image-set reasoning over 38 modalities and 18 departments), **OmniMedVQA** (Hu et al., 2024) (large-scale MCQ synthesized from 73 classification datasets), **ProbMed** (Yan et al., 2025) (adversarial reliability checks), and **SLAKE** (Liu et al., 2021) (physician-curated QA). To verify that SFT preserves general capabilities, we include **MMMU** (Yue et al., 2024) (general multimodal MCQ), **MultiMedQA** (Singhal et al., 2023a) (medical text-only MCQ), and **MMLU** (Hendrycks et al., 2021) (general text-only MCQ).

## Appendix D. Ablation Studies

All ablations experiments are conducted on *Aloe-Vision-7B* using the identical training configuration described in Table 6. Evaluation is performed following the protocol detail in Appendix C. For mixtures containing fewer samples than the one described in Section 3, we proportionally increase the number of training epochs to ensure that the model receives at least as many gradient updates as the baseline.

Table 7: Comparison of different data mixtures across all evaluation benchmarks. Accuracy (%) and LLM-as-judge score for SLAKE.

| Model | PathMMU | OmniMed | GMAI | ProbMed | SLAKE | MMMU | MultimedQA | MMLU |
|---|---|---|---|---|---|---|---|---|
| Final | 61.8 | 75.9 | 52.8 | 76.5 | 65.4 | 45.1 | 58.5 | 65.9 |
| Multimodal-only | 64.9 | 76.5 | 52.7 | 81.1 | 66.9 | 47.2 | 54.7 | 61.8 |
| Medical-only | 55.0 | 71.0 | 48.5 | 53.8 | 64.6 | 44.8 | 49.7 | 64.2 |

Table 8: With vs. without eval-image leakage. Accuracy (%) / SLAKE LLM-as-judge.

| Model | PathMMU | OmniMed | GMAI | ProbMed | SLAKE | MMMU | MultimedQA | MMLU |
|---|---|---|---|---|---|---|---|---|
| Without Leakage | 61.8 | 75.9 | 52.8 | 76.5 | 65.4 | 45.1 | 58.5 | 65.9 |
| With Leakage | 61.0 | 74.3 | 52.2 | 75.8 | 65.9 | 45.6 | 59.6 | 66.1 |

**Data mixtures.** We compare three compositions: (1) the final balanced mixture (Section 3), (2) a multimodal-only mixture (medical+general multimodal), and (3) a medical-only mixture (medical multimodal+medical text-only). Table 7 shows that medical-only underperforms consistently. The reason might be the higher percentage of medical text-only Chain-of-Thought samples (from 29.7% to 41.5% of loss tokens), which induces verbose outputs despite prompts requesting a single MCQ option. The multimodal-only model improves on several multimodal benchmarks but regresses on text-only tasks, indicating that general and medical text-only data are important to preserve language-domain competence.

**Evaluation leakage.** We measure the effect of removing evaluation images from training using exact image-hash matching (Section 3), which eliminates 6,273 samples ($\approx$0.18% of the pool). Two models are trained identically, one without leakage (excludes matches) and the other with leakage (includes them). As shown in Table 8, accuracy remains unchanged across benchmarks. We hypothesise that at this leakage rate, memorization effects are negligible for a 7B model, and matched images with potentially differing text can further attenuate any direct answer leakage.

**Filtering.** We test the effect of semi-automatic filtering (Section 3) by comparing training with vs. without removing low-quality samples. The non-filtered mixture contains 3,959,087 samples while the filtered mixture contains 3,571,622 samples (9.8% reduction). The non-filtered run executes 3,866 vs. 3,487 steps due to dataset's larger size. Performance is nearly identical overall (Table 9), with a notable gain on ProbMed (+2.4% absolute) for the filtered model. Filtering achieves comparable aggregate accuracy with ∼10% fewer samples.

## Appendix E. CareQA

### E.1. Results

Results for different models, organized by category (Medicine or Nursing) and by task type (multiple-choice or open-ended), are shown in Table 10. Across all models, performance on MCQ is consistently higher than on open-ended tasks, highlighting ongoing challenges in free-text medical reasoning. Larger models generally outperform smaller ones, with Aloe-Vision-72B achieving the highest MCQ scores and GLM-4.5V leading in the open

Table 9: Filtered vs. non-filtered mixtures. Accuracy (%) / SLAKE LLM-as-judge.

| Model | PathMMU | OmniMed | GMAI | ProbMed | SLAKE | MMMU | MultimedQA | MMLU |
|---|---|---|---|---|---|---|---|---|
| With Filtering | 61.8 | 75.9 | 52.8 | 76.5 | 65.4 | 45.1 | 58.5 | 65.9 |
| Without Filtering | 62.2 | 76.1 | 52.7 | 74.1 | 65.1 | 45.7 | 58.7 | 65.8 |

Table 10: Performance of different models on the CareQA-Vision dataset. Models highlighted in gray are general-purpose, while the others are domain-specific. Top table shows smaller models, and the bottom table shows larger models. Results are reported separately for multiple-choice (MCQ) and open-ended (VQA) formats, and further divided by category (Nursing and Medicine). Best results are shown in **bold**, and second-best results are underlined.

| Model | MCQ | | | VQA | | |
|---|---|---|---|---|---|---|
| | Overall | Nursing | Medicine | Overall | Nursing | Medicine |
| **Small models (<10B)** | | | | | | |
| Qwen2-VL-7B-Instruct | 51.52 | 35.71 | 56.91 | 26.47 | 16.07 | 29.17 |
| Qwen3-VL-8B-Instruct | 58.79 | 52.38 | 60.98 | 38.24 | **26.79** | 41.20 |
| MiMo-VL-7B-RL | 59.39 | 50.00 | 62.60 | 28.31 | 14.29 | 31.94 |
| Chiron-o1-8B | 47.27 | **57.14** | 43.90 | 21.32 | 17.86 | 22.22 |
| Lingshu-7B | 56.97 | 50.00 | 59.35 | 31.25 | 17.86 | 34.72 |
| HuatuoGPT-Vision-7B | 50.30 | 38.10 | 54.47 | 16.91 | 16.07 | 17.13 |
| Hulu-Med-7B | 59.39 | 40.48 | 65.85 | **43.38** | 23.21 | **48.61** |
| Aloe-Vision-7B | 56.36 | 38.10 | 62.60 | 36.76 | **26.79** | 39.35 |
| Aloe-Vision-7B-AR | **60.61** | 35.71 | **69.11** | 36.40 | 21.43 | 40.28 |
| **Large models (>10B)** | | | | | | |
| Qwen2-VL-72B-Instruct | 72.73 | 64.29 | 75.61 | 45.59 | 32.14 | 49.07 |
| Qwen3-VL-30B-A3B-Instruct | 71.52 | 57.14 | 76.42 | 43.75 | 23.21 | 49.07 |
| Kimi-VL-A3B-Instruct | 55.15 | 47.62 | 57.72 | 40.44 | 32.14 | 42.59 |
| GLM-4.5V | 72.73 | 66.67 | 74.80 | **64.71** | **51.79** | **68.06** |
| HuatuoGPT-Vision-34B | 54.55 | 47.62 | 56.91 | 24.26 | 16.07 | 26.39 |
| Lingshu-32B | 64.85 | 52.38 | 69.11 | 42.28 | 19.64 | 48.15 |
| Hulu-Med-32B | 63.64 | 38.10 | 72.36 | 48.53 | 28.57 | 53.70 |
| Aloe-Vision-72B | **77.58** | **69.05** | 80.49 | 49.63 | 37.50 | 52.78 |
| Aloe-Vision-72B-AR | 75.76 | 59.52 | **81.30** | 48.53 | 32.14 | 52.78 |

format. Notably, medicine questions are answered more accurately than nursing questions, suggesting uneven domain coverage in training data.

## E.2. Expert evaluation

As described in the main paper, experts labeled each model response as *correct, partially correct or incorrect* (see the interface in Figure 5). A response was considered *correct* when it matched the reference answer, even if the level of detail varies. It was labeled *partially*

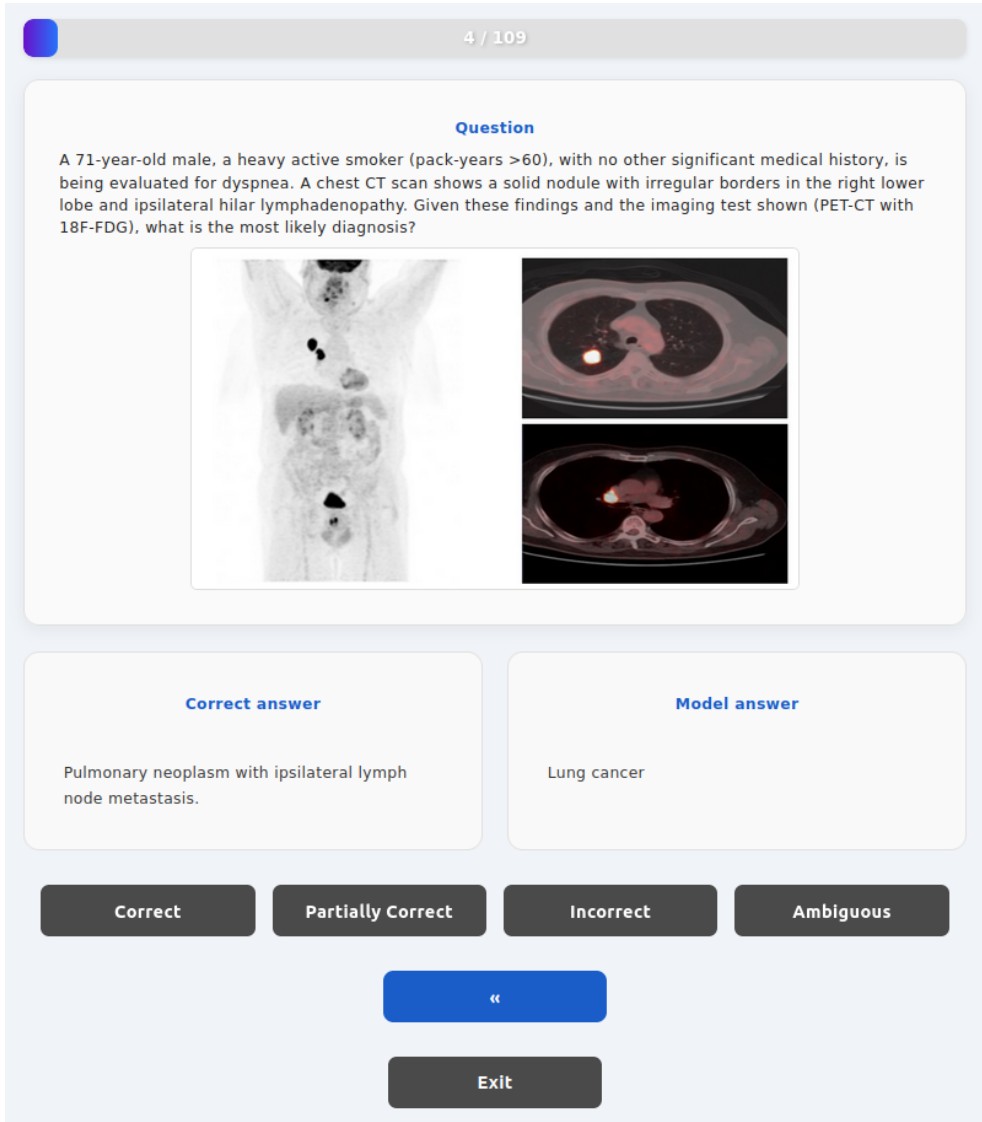

Figure 5: Interface used by experts to evaluate the model's answers.

*correct* when it captured only part of the required information, and *incorrect* when it failed to match the correct content.

Because the evaluation was performed on a dataset we created (CareQA-Vision), we introduced an additional label, *ambiguous*, to identify samples that were unclear, for example, poorly rephrased questions where multiple answers could be considered valid. This label was used exclusively for data cleaning: a sample was removed from the dataset if at least two experts marked it as ambiguous. Only three questions met this criterion, and were removed from the dataset. The remaining 105 questions from the medical CareQA-Vision were evaluated by four medical experts, with each question receiving at least two and up to four independent assessments (not all experts annotated all questions).

