# OpenReview forum: "Aloe-Vision: Robust Vision-Language Models for Healthcare"
_MIDL.io/2026/Conference — MIDL 2026 Poster_

### Official Review · Reviewer_EZii · 2025-12-16

**Confidence:** 4
**Preliminary Rating:** 4
**Final Rating:** 4

**Summary:**

This work presents a suite of open-source medical vision-language models with two scales (7B and 72B). In addition to these standard models, the authors provide robust versions that are trained on adversarial samples, improving resilience against adversarial attacks.

The authors curated a large-scale dataset, named Aloe-Vision-Data, which integrates data from both medical and general domains. This dataset is designed to enhance the models’ capacity to understand medical imagery while maintaining grounding in general visual concepts. Importantly, Aloe-Vision-Data is made publicly available as part of the open-source release.

The study also introduces a new benchmark for medical visual question answering (VQA) from Spanish residency exams.

**Strengths:**

- The paper provides an extensive review of related literature and includes broad comparative evaluations, clearly situating the proposed approach within the existing body of work.
- The authors demonstrate a strong commitment to openness and reproducibility by releasing the dataset, model checkpoints, and training recipes as open-source resources.
- Significant effort is dedicated to dataset curation, with careful consideration of data quality and diversity throughout the construction process.
- The curated dataset covers across multiple medical imaging modalities, supporting more comprehensive model training and evaluation.

**Weaknesses:**

- The paper would benefit from additional justification and deeper discussion of the experimental results, particularly in cases where performance trends are not immediately intuitive. For example, the Hulu-Med variants outperform larger models despite having substantially fewer parameters (32B vs. 72B), yet this behavior is not sufficiently analyzed or explained.
- Overall, the discussion of results is relatively limited, and key findings are often reported without adequate interpretation or contextualization. Given the availability of the additional two pages, the authors could expand the analysis of these observations, which would significantly strengthen the empirical arguments and improve the clarity of the conclusions.

**Detailed Comments:**

No minor comments, clear writing

**Justification Of Final Rating:**

I think the paper is a good contribution to the field, and I support open-source works which should encourage other people in the community, given the data scarcity problems compared to general computer vision domain.

**Justification Of The Preliminary Rating:**

The paper is technically very solid, and the open-source release of the models, data, and training details is a strong contribution for the community that significantly enhances reproducibility and practical impact.

**Questions To Address In The Rebuttal:**

Please see weaknesses, with a deeper and better discussion of the results, I am keen to increase my rating

---

> ### Author Response · Authors · 2026-01-25
>
> We thank the reviewer for this constructive feedback. While the original manuscript was constrained by space limitations, we agree that several performance trends require deeper analysis. In the revised version, we have expanded the results section to provide additional interpretation and contextualization of the observed behaviors.
>
> In particular, we now discuss the seemingly counterintuitive observation that Hulu-Med-32B achieves performance comparable to or exceeding that of larger models such as Aloe-Vision-72B-AR. This behavior can be explained by a combination of architectural, data, and training-procedure differences rather than parameter count alone. From an architectural standpoint, Hulu-Med is built on the more recent Qwen-2.5 backbone and incorporates a custom vision encoder, whereas Aloe-Vision is based on the Qwen2-VL architecture. From a data perspective, Hulu-Med is trained on a large-scale medical dataset comprising approximately 16.5M samples. Finally, the training pipelines differ significantly, Hulu-Med follows a three-stage procedure involving vision encoder and projector alignment, continuous pretraining, and fine-tuning, while Aloe-Vision adopts a single-stage supervised fine-tuning strategy.
>
> Taken together, these factors suggest that model scale alone is not a reliable predictor of performance in medical vision-language tasks. Instead, architectural choices, dataset scale, and training curriculum can significantly influence downstream performance.  As future work, a systematic study disentangling the relative contributions of model size, data scale, and training strategy would be valuable to better understand these trade-offs.

---

> > ### Comment · Reviewer_EZii · 2026-01-26
> >
> > Thanks for the comments and revision of discussion part. I believe this work is a contribution and a value to the community, and seeing these kind of work in conference should encourage more people to open source resources.

---

### Official Review · Reviewer_QwwK · 2026-01-04

**Confidence:** 5
**Preliminary Rating:** 2
**Final Rating:** 4

**Summary:**

This paper introduces Aloe-Vision, a family of open-source medical LVLMs. The authors build these models by finetuning on a curated dataset (their Aloe-Vision-Data) that mixes medical and general domain data. To test performance, they evaluate on various benchmarks and against other general or medical domain LVLMs. They also release CareQA-Vision, a new benchmark derived from Spanish medical residency exams. Finally, the study emphasizes safety by incorporating adversarial training.

**Strengths:**

- There is a good mix and variety of baseline models in the study, which makes for a good comparison against previous models (both general and domain-specific).
- Similarly, the work selects a good number of various benchmarks to assess the performance of their own model against the baselines.
- Overall, the writing and structure/format of the paper are good. Is it easy to read and follow.

**Weaknesses:**

- The contributions are a bit overblown. The main contribution here is the data curation (which I would argue is not a big enough contribution). The models themselves are simply Qwen-VL base models finetuned on this new curated dataset. The new proposed benchmark is also too small to be useful in practice.
- There are methodological issues in the usage of LLM-as-a-judge due to using the same model family (Qwen-VL) for quality score assessments (see detailed comments for more).
- I am not sure how impactful or useful this work could be. The proposed models are not SOTA and often underperform compared to other baselines. In practice, medical experts would prefer a very strong model on their specific task rather than an average one on all tasks. This again undermines the main contribution of the paper, which is mostly the dataset curation.
- See detailed comments, where I explain all of these points.

**Detailed Comments:**

- It would have been nice to see one or two closed-source LVLMs (GPT4V, Gemini, ...) in the list of baselines, as a reference to get an idea of the higher performance bound. I am aware that the authors focus here on open-source models, and it is one of the motivations of the paper, but it is always good to have one of these as an upper-bound performance.
- Lack of novelty on the model/architecture side: There is no novelty here on the model/architecture side. The authors simply replicate the training recipes of Qwen-VL on their own dataset. The authors need to clarify that the model itself is not really a contribution.
- I am not sure about the usefulness and applicability of the suite of models trained and provided by the authors here. In most benchmarks, they are not SOTA (often underperforming compared to other baselines). They may offer a more balanced performance on average, but in practice, a practitioner will always prefer the best model for their specific task/needs rather than a good average model. The authors need to elaborate on the usefulness and impact of these kinds of models.
-  There is a major issue in this work: the authors often use Qwen-VL variants to assist in curation and quality score assessment. A well-documented issue in LLM-as-a-judge pipelines is something called self-enhancement bias (favouring their own answers). A good practice is to use a different model architecture as a judge (ideally, a very strong model such as GPT5 or Claude). The authors need to address this issue.
- Although a new proposed benchmark is a good idea, I would not consider performance on the new benchmark provided by the authors as a good indicator of generalization or performance since it is super small. Also, I am not convinced about the generation of this benchmark. The authors mention that they translate the questions and check their validity, but translation of medical terms is often delicate and needs very careful quality assessment.
- Assessing LLM-as-a-judge's validity by comparing to human evaluation is a good practice, but as the authors point out, agreement is not very high, putting into question the validity of the whole LLM-as-a-judge method here (especially the choice of judge).
- The adversarial training section uses only one adversarial benchmark (HEART), which is very limited and not enough to draw any general conclusions.
- Performance of the authors' own models is a bit overblown, often underperforming compared to at least one or two baselines.

**Justification Of Final Rating:**

I thank the authors for making a lot of effort to address my criticisms and improve the paper. I decided to increase my rating to weak accept, as most of my comments were addressed. I particularly appreciate the precision added to the contributions of this work, the addition of a closed model (GPT 5.2) in the results as a reference, and the discussion on the LLM-as-a-judge biases.

**Justification Of The Preliminary Rating:**

I go with a weak reject here. Mainly, the only contribution of this work, in my opinion, is the dataset curation. The model itself is just a finetuned Qwen-VL on this new data, and the new benchmark is extremely small to be useful (130 questions only). Moreover, there are methodological issues, such as the LLM-as-a-judge approach. The performance of their models is also underwhelming, so it is hard to motivate their usage in practical applications.

**Questions To Address In The Rebuttal:**

The three main points to address would be:
- Clarifying the contribution of this work. The contribution here is simply the dataset curation.
- Addressing the LLM-as-a-judge issue of using the same type of model for quality score assessment. Could the authors improve this?
- Performance of the new finetuned models proposed is underwhelming. I struggle to see how these would be useful in practice, considering practitioners would always prefer the best model for a given task rather than an average model (like the ones the authors propose here).

Ideally, all the points in the detailed comments should be addressed.

---

> ### Author Response · Authors · 2026-01-25
>
> We thank the reviewer for the constructive feedback. We have revised the manuscript to clarify the scope and contributions of this work and to incorporate the reviewer's suggestions. We have also addressed all comments and questions in the response below.
>
> First, we would like to make the main contribution of this paper clear. Our work does not introduce architectural novelty, its primary contribution lies in the curated dataset, benchmark and reproducible training framework. Regarding the usefulness of the released models, they are not intended for direct clinical deployment. Instead, they are designed to support research. Although they do not achieve state-of-the-art performance on every benchmark, they remain competitive overall while showcasing stronger robustness under adversarial conditions. We view these results as a step toward more reliable medical multimodal models. We have revised the manuscript to more accurately present the model's performance.
>
> Following the reviewer's suggestion, we are evaluating the benchmarks in Table 3 using GPT-5.2. This comparison helps contextualize the performance of open-source approaches and highlights the need for continued research in open-source models. Due to resource constraints, not all results are available at the time of the rebuttal, the remaining results, including the adversarial benchmark, will be added in the final version.
>
> We next address the concerns regarding the use of Qwen-VL for curation and quality score assessment, and clarify their limited role in the pipeline. The curation steps are intended to only remove clear low-quality or incoherent samples. As the reviewer points out, using Qwen-VL to score image-question-answer relatedness can reflect model specific preferences. To mitigate this, we use the quality score only for conservative filtering based on an arbitrary threshold. We discard the samples with worse quality, so the score does not drive any model optimization.  Despite that, there is still a risk where the curation model could favor synthetic samples generated by the same family.
>
> For evaluation, we first note a correction in the implementation of judge-human alignment score. Previously, Krippendorff's alpha was calculated treating all three classes as nominal, meaning all disagreements were treated equally, regardless of which categories were involved. Now, we treat the three categories as ordinal. With this fix, the agreement increases from 0.653 to 0.796, and alpha between human evaluators and LLMs to 0.812 indicating that the LLM judges are largely consistent with human expert assessments.
>
> Regardless, we acknowledge the risks of using a single judge that belongs to the same model family as some of the evaluated MVLMs, and include two additional judges: Llama-3.3-70B and Olmo-3-32B. We use majority voting among the three judges, which we found to have the highest agreement with humans, at around 80\% (more details in the revised paper). As a final remark, the majority of our benchmarks are still close-ended, only SLAKE and CareQA-V include open-ended questions (besides a MCQ split).
>
>
> Regarding the proposed benchmark size and construction, while it is small, it is generated from real medical examination questions designed and reviewed by medical experts. This ensures that the questions are clinically meaningful and do not contain incorrect or misleading information, an issue that has been observed in some larger automatically collected benchmarks. While we acknowledge medical translation is inherently delicate, no errors were identified by domain experts during evaluation. Furthermore, the benchmark provides a meaningful measure of generalization because the questions are highly unlikely to appear in existing training corpora. The source materials are distributed exclusively as non-English PDF documents and require substantial processing to be incorporated into large-scale datasets.
>
> We also made the adversarial evaluation setup clearer and more detailed. HEART is referred to as a single benchmark, but it is a dataset constructed from eight distinct medical datasets covering different tasks and imaging modalities. The adversarial modifications are applied independently to each of these subsets. Only one subset (FractAtlas) is used for adversarial fine-tuning, while the remaining seven datasets are used exclusively for evaluation, showing that the reported results reflect generalization to unseen data, regardless of the image modality. In total, the evaluation spans 13,645 samples, which we consider is sufficiently large and varied. We have expanded the manuscript to make the composition of HEART and the total evaluation size explicit.
>
> Overall, we hope these clarifications better communicate the central value of this work and we are happy to provide further detail during the discussion phase if any points remain a concern.

---

### Official Review · Reviewer_SjoC · 2026-01-16

**Confidence:** 3
**Preliminary Rating:** 5
**Final Rating:** 5

**Summary:**

This paper introduces Aloe-Vision, a family of open and reproducible medical vision-language models (LVLMs) designed to improve performance on clinically relevant multimodal tasks, such as open-ended medical question answering, interpretation, and diagnostic reasoning. To support training, the authors construct Aloe-Vision-Data, a large-scale, balanced, and quality-filtered instruction mixture spanning both modality (multimodal vs. text-only) and domain (medical vs. general). The paper further introduces CareQA-Vision, a contamination-free medical benchmark derived from Spanish residency examinations, which enables evaluation of generalization to previously unseen cases and open-ended outputs. In addition, the robustness-enhanced variant Aloe-Vision-AR demonstrates that explicit robustness training mitigates vulnerability to misleading or sycophantic prompts, achieving strong performance on the HEART adversarial benchmark. Finally, the paper employs an LLM-as-a-judge evaluation protocol and validates its reliability via expert comparison (62.86% agreement with expert consensus), showing that Aloe-Vision matches or surpasses state-of-the-art LVLMs across 10 standard benchmarks.

**Strengths:**

The paper presents several notable strengths and valuable contributions to the medical vision-language modeling community. First, the construction of Aloe-Vision-Data provides a high-quality and large-scale training mixture spanning both modality (multimodal vs. text-only) and domain (medical vs. general), which is non-trivial in a field constrained by data scarcity, heterogeneity, and annotation cost. Second, the introduction of CareQA-Vision yields a contamination-free benchmark with both MCQ and open-ended formats, covering multiple specialties and imaging modalities in a manner that better approximates real clinical reasoning and diagnostic interpretation tasks compared to prior simplified VQA setups. The open release of data, benchmarks, and model weights further enhances community value and is likely to catalyze follow-up research in medical LVLMs.

In addition, the paper systematically evaluates robustness under adversarial and misleading prompt conditions, addressing safety-critical failure modes such as sycophancy that are highly relevant for medical deployment. Finally, the experimental evaluation is comprehensive, spanning multiple datasets, task formats, and judging protocols, including validation of the LLM-as-a-judge framework with human experts.

Overall, it provides meaningful resources and insights that should benefit future research in medical LVLMs.

**Weaknesses:**

The paper’s primary contributions lie in the construction of datasets and evaluation resources, while the methodological novelty is relatively limited. The training approach mainly integrates existing data sources and applies single-stage supervised fine-tuning on a pre-trained Qwen2-VL-Instruct backbone, making the work more of a benchmark/resource contribution than a new learning algorithm. This is not necessarily negative given the practical importance in medical applications, but it should be clearly positioned as such.

Additionally, although the experiments compare against strong general-purpose LVLMs, the evaluation does not include specialist medical models tailored for specific domains (e.g., radiology or ophthalmology systems), which could provide useful context for assessing performance trade-offs between broad medical LVLMs and domain-specific solutions. Including such discussion or justification would strengthen the paper’s positioning and clarify deployment scenarios.

**Detailed Comments:**

I have no additional comments. The paper is well written, the contributions are clearly articulated, and I found no issues that require clarification beyond what is already presented.

**Justification Of Final Rating:**

I recommend acceptance because the paper makes meaningful and timely contributions to the medical vision-language modeling community through the release of a high-quality training mixture, a clinically relevant benchmark, and reproducible model artifacts. While the methodological novelty is modest compared to the benchmark and evaluation contributions, the demonstrated robustness under adversarial prompts, the comprehensive experimental validation, and the overall clarity and completeness of the presentation collectively justify a positive assessment. I expect the released resources to have practical impact and enable further research in medical LVLMs.

**Justification Of The Preliminary Rating:**

I recommend acceptance because the paper makes meaningful and timely contributions to the medical vision-language modeling community through the release of a high-quality training mixture, a clinically relevant benchmark, and reproducible model artifacts. While the methodological novelty is modest compared to the benchmark and evaluation contributions, the demonstrated robustness under adversarial prompts, the comprehensive experimental validation, and the overall clarity and completeness of the presentation collectively justify a positive assessment. I expect the released resources to have practical impact and enable further research in medical LVLMs.

**Questions To Address In The Rebuttal:**

I have no questions for the authors to address in the rebuttal.

---

> ### Author Response · Authors · 2026-01-25
>
> We thank the reviewer for their thorough and positive evaluation of our work. We appreciate their assessment of the contributions, experimental validation, and clarity of presentation, as well as their recognition of the practical impact of the released resources. We are grateful for the constructive feedback and encouragement.

---

### Author Rebuttal · Authors · 2026-01-25

**Rebuttal:**

We thank all reviewers for their careful reading of the manuscript and for the constructive and insightful feedback. In response, we have revised the paper to improve clarity, scope definition, and depth of analysis. In particular, we (i) clarified the primary contributions of this work, emphasizing the dataset, benchmark, and reproducible training framework rather than architectural novelty, (ii) extended the presentation and interpretation of experimental results, including additional discussion of observed performance trends, (iii) strengthened the evaluation methodology by correcting the judge-human agreement calculation and incorporating multiple LLM judges, (iv) expanded and clarified the description of the benchmark construction and adversarial evaluation setup to better highlight generalization and robustness, and (v) extended the evaluation to include GPT-5.2 as a representative closed-source state-of-the-art LVLM. Some GPT-5.2 results are not yet available at the time of rebuttal and the missing accuracies will be added in the final version of the manuscript. We are grateful for the reviewers’ suggestions, which helped strengthen the final manuscript.

**Supporting Material:**

/attachment/669c9037919551909e28dfca1db82dff580d9ae3.pdf

---

### Meta-Review · Area_Chair_dviM · 2026-02-10

**Recommendation:** Accept (Oral)
**Confidence:** 5

**Metareview:**

The reviewers reached a consensus for acceptance, praising the work's meaningful contributions to reproducibility in medical vision-language modeling through the release of training data, benchmarks, and model checkpoints. The rebuttal successfully addressed initial concerns about evaluation methods, further strengthening the assessment that these resources will have strong practical impact on the field.

---

### Decision · Program_Chairs · 2026-02-13

Accept (Poster)